# Trypsin and Trypsinogen Activation Peptide in the Prediction of Severity of Acute Pancreatitis

**DOI:** 10.3390/life14091055

**Published:** 2024-08-23

**Authors:** Andreas Allemann, Sebastian M. Staubli, Christian A. Nebiker

**Affiliations:** 1Department of Psychiatry Biel, PZM AG, Vogelsang 84, 2501 Biel, Switzerland; 2HPB and Liver Transplantation Service, Royal Free London NHS Foundation Trust, Pond Street, London NW3 QG, UK; 3Department of Surgery, Cantonal Hospital Aarau, Tellstrasse 25, 5001 Aarau, Switzerland

**Keywords:** trypsinogen, trypsin, TAP, acute pancreatitis, severity

## Abstract

Objectives: To assess the predictive value of serum trypsin and trypsinogen activation peptide (TAP) for the severity of AP through a single center cohort study as well as a systematic review of the current literature. Methods: A literature search was conducted using Medline (PubMed), EMBASE and the Cochrane Central Register. A total of 142 patients with acute pancreatitis (AP) were included in the cohort study and parameters of the revised Atlanta criteria of 2012 and the APACHE II were assessed. Results: The review showed promising results for the predictive value of serum trypsinogen-2 but conflicting results for serum TAP and trypsin. In the cohort study, patients were observed for 4 days after diagnosis of AP; 9 patients had severe AP, 35 patients had moderate AP and 81 patients had mild AP. The ratio of the geometric mean of severe vs. mild AP for trypsin was 0.72 (95% CI: 0.51–1.00), *p* = 0.053 and, for TAP, 0.74 (95% CI: 0.54–1.01), *p* = 0.055, respectively. Conclusions: The cohort study showed an inverse correlation of serum levels of TAP and trypsin with severity of AP. Serum TAP and trypsin have an inferior predictive value of severity of AP compared to the clinical APACHE II score.

## 1. Introduction

Acute pancreatitis (AP) is an inflammatory disease, frequently caused by gallstones or excess alcohol consumption [1]. Its course is highly variable and ranges from mild to severe. According to the revised Atlanta Classification (rAC) of 2012 [2], there are three degrees of severity. Mild (MAP), which is self-limiting, is characterized by the absence of organ failure and local complications and carries a low mortality and usually resolves spontaneously. Moderately severe acute pancreatitis (MSAP) is characterized by transient organ failure (less than 48 h duration) and/or local or systemic complications. In about 20% of cases, patients suffer from severe acute pancreatitis (SAP), characterized by persistent organ failure (lasting more than 48 h) often requiring intensive care and with an associated mortality of up to 30% if infected necrosis occurs [3]. Multiorgan failure (MOF) and necrosis are the main causes of morbidity and mortality in AP [4].

A variety of scores have been developed in the past to predict or assess the severity of AP based on the shift in laboratory markers over 48 h (e.g., Ranson score) [5], on CT images (e.g., Balthazar score) [6], or on clinical and laboratory findings such as the acute physiology and chronic health evaluation score (APACHE II) [7]. The current predictive models lack specificity and reliability, especially in the early course of disease [8]. In contrast, the revised Atlanta classification (rAC) places more emphasis on organ dysfunction (as defined by the modified Marshall score [2]). Patients with SAP should be recognized early for monitoring and supportive treatment, for example, fluid resuscitation [9]. These patients should also be considered for treatment in an intensive care unit [10]. A large number of laboratory parameters have been examined to predict the severity of AP. C-reactive protein (CRP) remains the most widely used marker with a cut-off of 150 mg/L at 48 h [11]. Amongst other parameters, our group was previously able to demonstrate that cortisol [12] is a valuable predictor of development of organ failure and death [13]. An ideal specific biomarker to reliably assess and predict the severity of AP has not yet been identified. In this study, we focus on the role of trypsinogen, trypsin and TAP as predictors of severity in AP. These markers have been studied previously, with conflicting results [14,15,16,17].

The initial phase of AP occurs within the acinar cells of the pancreas through activation of pancreatic proenzymes. The activated pancreatic enzymes leak into pancreatic tissue, which, in turn, leads to autodigestion, causing edema, apoptosis, hemorrhage and cellular lysis [4], resulting in a complex sequence of inflammatory reactions, which determine the further course of the disease [18].

Under physiological conditions, the proenzyme trypsinogen is activated by cleavage of trypsinogen activation peptide (TAP) through the duodenal brush-border enzyme enterokinase resulting in trypsin and TAP (see Figure 1) [19]. In AP, intra-acinar activation of trypsinogen leads to acinar cell injury. In addition, activated trypsin is able to cleave TAP from trypsinogen, leading to further autoactivation of trypsin [18,20].

There are two major human isoenzymes of trypsinogen: trypsinogen-1 (cationic trypsinogen) and trypsinogen-2 (anionic trypsinogen). In the bloodstream, trypsin is rapidly inactivated by α_2_-macroglobulin and α_1_-antitrypsin (AAT) [21]. In humans, cleavage of trypsinogen-1 or trypsinogen-2 results in the same activation peptide (TAP) in equimolar quantity as trypsin [19]. Trypsin is a member of the serine peptidase family that cleaves peptide bonds [22].

Part of early pancreatic damage during AP is due to excess trypsin activation and, therefore, serum trypsin levels, as well as serum TAP levels, may correlate earlier in the course of the disease with its severity than nonpancreatic specific markers. There are various publications examining the correlation between serum trypsinogen-2 and the severity of AP. Some showed promising results [14,23,24], whereas others concluded no discriminatory value of trypsinogen-2 between mild and severe AP [25]. The aim of this study is to present the results of our single-center observational study and provide a framework to interpret these data by providing a systematic review of the literature of the studied biomarkers.

## 2. Single-Center Observational Study

### 2.1. Methods

#### 2.1.1. Study Design and Population

This study was designed as a single-center, observational cohort study at the University Hospital of Basel, Switzerland. The cohort had been studied previously by our group [12]. The study protocol was approved by the local ethics committee (Ethikkommission beider Basel, EKBB 281/10) and registered on ClinicalTrials.gov (NCT01293318). The methodology was performed according to all relevant guidelines and regulations. All information about the patient’s identity and clinical data remained on the hospital’s password-protected server. All patient data were anonymized. From April 2011 to January 2015, all patients diagnosed with AP admitted to the University Hospital Basel, Switzerland, were eligible for the observational study, following diagnosis of AP either in the emergency department or on the ward.

The inclusion criteria were informed written consent, age older than 18 years, and less than 96 h between the onset of abdominal pain and study inclusion. Pregnancy was not an exclusion criterion. Initial treatment followed local emergency standards of care (www.emergencystandards.com (accessed on 1 January2011)) [12].

#### 2.1.2. Clinical Assessment

On study inclusion, medical personnel recorded the vital parameters and filled a short questionnaire assessing baseline characteristics. Routine blood samples, as well as two additional tubes (1× serum and 1× EDTA) of 7.5 mL each, intended for this study, were drawn at admission and at day 2. The required parameters of the rAC of 2012, as well as the Ranson, APACHE II and the modified Marshall scores, were collected from electronic health records and noted on case report forms. Patients were observed during the first 4 days after the diagnosis of AP. Mortality and computed tomography scans were evaluated in patients throughout the duration of hospitalization. Patients discharged within 4 days of admission remained in the study irrespective of incomplete follow-up data. If blood oxygen saturation was higher than 90%, arterial blood gas analysis was omitted and a partial pressure for oxygen of 90 mmHg was assumed and used for further calculation of scores. The severity of pancreatitis was classified according to the rAC over the 4-day observation period. Local complications were defined as peripancreatic fluid collection, pancreatic and peripancreatic necrosis, pseudocyst and walled-off necrosis. Organ failure was defined according to the modified Marshall score and the rAC criteria [12].

#### 2.1.3. Assay

The blood samples drawn for the study were centrifuged and immediately frozen at −80 °C. EDTA plasma levels of trypsin were determined using the commercially available human Trypsin ELISA kit (SEA230Hu, Cloud-Clone Corp, Houston, TX, USA). Plasma levels of TAP were determined using the commercially available human TAP ELISA kit (CEA634Hu, Cloud-Clone Corp, Houston, TX, USA) assay. These kits fulfilled the required coefficients of variation. The limits of detection of the mentioned assays are 15.6–1000 pg/mL and 123.5–10,000 pg/mL, respectively [12].

#### 2.1.4. Sample Size

A formal sample size calculation was performed on the basis of the primary research question of our previous study [12]. The present study is exploratory in nature and is based on the available data for the cohort of patients with AP.

#### 2.1.5. Statistical Analysis

We used linear regression models to assess the associations of trypsin and TAP with disease severity according to the rAC criteria. Biomarker levels were analyzed by logarithmic transformation. The results are presented as estimated ratios of geometric means (with their 95% confidence intervals).

#### 2.1.6. Development and Assessment of Prognostic Models

To assess the prognostic accuracy of trypsin, TAP and the APACHE II score as measured on admission (day 0) in predicting the combined endpoint of organ failure or death within the first four days after study inclusion, we used logistic regression models and calculated the area under the receiver operating curve (AUC) separately for each prognostic variable. In addition, the results are presented as estimated odds ratios (with their 95% confidence intervals) per standard deviation increase in the log-transformed prognostic variables [12].

#### 2.1.7. Statistical Software

For the analyses and graphics, we used R 3.2.1 (R Foundation for Statistical Computing, Vienna, Austria) and, for the ROC analysis and reclassification methods, we used the ROCR and PredictABEL add-on packages [26].

## 3. Results

### 3.1. Patients

This post hoc analysis of a prospective, single-center, observational cohort study included 142 patients. Seven patients had two episodes of AP during the study period and, in these cases, only the first episode was considered. The clinical and demographic characteristics are shown in Table 1. The median age was 57 years, 43% were female, 61% presented with a biliary etiology of AP, and 18% had a history of AP without criteria for chronic pancreatitis. A total of 40% of patients underwent a CT scan.

### 3.2. Severity of AP and Biomarker Serum Levels

According to the rAC, 9 patients were classified as having severe AP, 35 as having moderate AP, and 81 patients as having mild AP. In 17 patients, missing covariate information precluded the classification of disease severity. Baseline characteristics and biomarker levels are shown in Table 1. Overall, patients with severe disease had lower serum TAP levels at study inclusion than patients with mild disease. The same tendency was observed with serum trypsin levels in patients with severe vs. mild disease see Figure 2.

In a sensitivity analysis, we excluded a few outliers with unusual and potentially confounding values but, generally, the results were compatible with our main analysis, as shown in Table 2.

### 3.3. Prediction of Organ Failure and Death

Twenty percent of all patients had organ failure according to the modified Marshall score upon study inclusion or within four days after admission. Five patients died during hospitalization. Assuming that the 10 patients discharged before the end of the four-day observation period neither died nor developed organ failure within four days following study inclusion, a total of 30 patients died or developed organ failure within four days of admission.

Trypsin, as well as TAP serum levels, showed an inverse correlation with the APACHE II score and presence of organ failure or death. In SAP, trypsin and TAP levels tend to be lower than in patients with mild AP. The predictive ability of the biomarkers was weak and inferior to the APACHE II score (odds ratio (OR) of 2.76 and AUC of 0.73 (*p* < 0.001)). The OR for trypsin is 0.64, with an AUC of 0.66 (*p* < 0.036). The OR for TAP is 0.88, with an AUC of 0.55 (*p* < 0.575) (Table 3) (Appendix A).

## 4. Systematic Review

### 4.1. Study Identification, Eligibility and Screening

A systematic review of articles published in English using Medline (PubMed), EMBASE and the Cochrane Central Register was conducted to identify eligible articles. The systematic review was conducted in accordance with the Preferred Reporting Items for Systematic Reviews and Meta-analysis (PRISMA) statement [27]. The terms “acute pancreatitis”, “trypsinogen”, “trypsin”, “TAP”, “serum”, “plasma” and “severity” were used in different combinations for the search. The following search strings were used “acute pancreatitis trypsin serum severity, acute pancreatitis trypsinogen severity, acute pancreatitis TAP severity serum, acute pancreatitis TAP severity plasma, acute pancreatitis trypsin prospective study, acute pancreatitis trypsinogen serum, acute pancreatitis trypsinogen plasma severity”. Eligibility criteria consisted of (1) clinical studies published in peer-reviewed English-language journals, with no restriction as to year of publication, (2) either serum levels of trypsinogen-1, trypinogen-2, TAP or trypsin were assessed in patients with AP, and (3) serum levels of the aforementioned markers were put in relation to the severity of AP. Studies using assays that were unable to differentiate between trypsinogen-2 and trypsin in complex with α_1_-protease inhibitor (immunoreactive trypsinogen-2) were excluded. Two searches were performed by two investigators (AA and SMS). Once the database search had been conducted and eligible publications had been identified, the reference lists of the selected publications were hand searched in addition.

### 4.2. Data Extraction and Quality Assessment

Regarding the data extraction, we placed emphasis on the area under the curve (AUC) values measured by the receiver-operating characteristics (ROC) curve analysis for the prognosis of AP (severe vs. mild), ideally at admission or within 72 h of symptom onset. As AUC values were not available for every study, we also listed the absolute median concentrations of the respective markers.

The systematic review provides a qualitative analysis of whether serum levels of trypsin, trypsinogen-1, trypsinogen-2 and TAP on admission are useful in clinical practice as prognostic markers of the severity of AP. The eligible studies included consecutive patients presenting with AP in a defined time period. Demographic and etiologic information are presented. Studies focusing on post-ERCP-induced AP only were excluded. Patients with abdominal pain of extra-pancreatic origin and no history of AP drawn from the same patient population usually served as controls. Patients were classified prospectively according to the AC or rAC. In one publication, patients were classified according to a grading system of AP that was not further specified [28].

## 5. Findings

The literature search yielded a total of 970 results, excluding duplicates in different search strings, and an additional search of the reference lists yielded a further 6 eligible publications. From the total sum of publications, 907 were excluded by title screening using the previously mentioned criteria. Of the remaining 69 publications, 40 were excluded by abstract screening. Of the remaining 29 publications 17 were excluded following assessment of the full text and, thus, 12 studies were included in the final analysis. Of the six publications extracted from the references, one relevant publication was included [17] (see Figure 3).

### 5.1. Trypsinogen

Seven publications examining the correlation between serum levels of trypsinogen and severity of AP, with a total of 828 patients, were included (Table 4).

The most recent publication, also the one with most AP patients included (N = 239), showed that serum levels of trypsinogen-2 on admission (AUC 0.726) outperformed trypsinogen-1 (AUC 0.656) [29] in predicting the development of organ dysfunction and a severe course of disease.

Three publications by Hedström et al. examined serum levels of trypsinogen-2 in connection with the diagnosis and prediction of severity of AP. In these studies, an AUC of 0.744 for trypsinogen-2, for the differentiation between a mild and severe course of disease 24 h after admission, was reported [15]. A publication comparing levels of urinary and serological trypsinogen-2 found an AUC of 0.721 for the differentiation of mild vs. severe AP on admission for serum trypsinogen-2, performing similarly to levels of urinary trypsinogen-2 [30]. Another study by the same group examining levels of trypsinogen-2 in the serum in AP found an AUC of 0.792 for the differentiation of a mild vs. severe course of disease, performing significantly better than lipase and CRP levels [14]. Lempinen et al. found that an AUC of 0.745 at 24 h after admission for serum levels of trypsinogen-2 could differentiate between a mild and severe course of AP, performing similarly to trypsinogen-1 (AUC of 0.768) but significantly poorer than urinary levels of trypsinogen-2, with an AUC of 0.925 [24]. A study by Regnér et al. showed significant discriminatory ability between mild and severe disease by determination of serum trypsinogen-2 levels, and the time course analysis of trypsinogen-2 levels showed significantly higher levels on the first 3 days of hospitalization in severe vs. mild disease [31]. Another time course analysis of serum levels of trypsinogen-2 by Kemppainen et al. showed a significant discrimination between mild and severe disease, most pronounced on admission and gradually declining thereafter.

### 5.2. TAP

We found four publications examining the correlation of serum levels of TAP with severity of AP, including a total of 295 patients (Table 5). A study by Mayer et al. with 25 patients, 16 of whom presented with SAP according to rAC, found serum levels of TAP at admission could significantly differentiate between a mild and a severe course of disease [32]. Similarly, Kemppainen et al. showed a statistically significant difference in serum levels of TAP at admission between patients with a mild vs. a severe course of disease. Thereafter, serum TAP levels declined rapidly to below the limit of detection [17]. Lempinen et al. only found elevated serum TAP levels in severe AP on admission, followed by a rapid decrease to undetectable levels, whereas, in mild AP, the concentrations remained mostly under the detection level of the assay. On admission, the AUC for differentiating between a mild and a severe course of disease was high at 0.823 [24]. Pezzilli et al. found no significant predictive value of serum TAP levels, which were under the limit of detection in 70.6% of patients with AP. In a time-course analysis the serum concentrations of TAP were slightly higher in patients with MAP than SAP, although this was not statistically significant. No significant difference in serum TAP levels was found in patients with AP compared to patients with acute abdominal pain of extra-pancreatic origin [16].

### 5.3. Trypsin

We found one publication examining the correlation between serum levels of trypsin and severity of AP, with a total of 140 patients (Table 6). Hu et al. studied serum trypsin levels in acute pancreatitis, showing increased levels in patients with SAP (grade III–IV) vs. MAP (grade I–II) according to the Determinant-Based Classification of AP [28].

## 6. Discussion

Our study suggests that trypsin and TAP are inferior to the APACHE II score in predicting the severity of AP. Our literature review showed promising results for predicting the severity of AP by assessing serum levels of trypsinogen-2 on admission. Studies investigating the predictive value of trypsinogen-2 for the assessment of severity of AP often showed a two-fold increase in median serum concentrations exceeding in SAP vs. MAP—a finding that translates into promising AUC values in most of the publications. In our own cohort, however, we could not confirm these results. Both serum parameters, trypsin and TAP, were outperformed by the APACHE II score as measured by AUC (0.66, 0.55 and 0.73, respectively).

The strength of the study lies in the high number of enrolled patients, in the use of the latest classification system (rAC) and in the prospectively collected mostly complete outcome data regarding a population of AP patients with all etiologies. One possible weakness of our clinical study is the low ratio of cases with SAP vs. MAP compared to other publications. One possible explanation might be the rather short observation period of 4 days, as sometimes SAP occurs later in the course of disease. This is not a prospective clinical trial but a post hoc analysis of prospectively collected data. Patients excluded were not included in a separate intention to diagnose analysis. Patients with an onset of symptoms of up to 96 h before admission were included and peak levels of markers that decrease over time, e.g., TAP, might have been missed or underestimated. Ten patients were discharged due to an uneventful recovery before completion of the 4-day observation period. Trypsinogen-2 is the most promising marker according to our systematic review, as well as the one with the most clinical data available. In our clinical study, we did not assess the predictive value of trypsinogen-2 for the severity of AP. Clinical studies on serum TAP and trypsin for the prediction of AP are sparse, and the number of patients enrolled are often low and present conflicting results. Another factor is the consideration of the references from the selected publications literature and subsequent review and inclusion of one publication not detected through our database searches.

Our clinical study shows an inverse correlation between serum levels of TAP and trypsin and the severity of AP. Patients with a severe course of AP show lower levels of trypsin and TAP at study inclusion than patients with a mild course of disease. Other publications show higher serum levels of TAP and trypsin in patients with SAP than MAP. The studies examining serum TAP levels in AP describe low serum levels of TAP at admission, sometimes even below the limit of detection, and, if detectable, rapidly declining thereafter [16,17,24], limiting its usefulness in clinical practice. As TAP is a relatively small molecule of 7–10 amino acids, it is rapidly eliminated from the circulation by renal excretion, with a half-life of about 8 min [33]. Therefore, the publications by Tenner et al. [34] and Neoptolemos et al. [35] focus on urinary TAP concentrations.

Results pertaining to the serum levels of trypsin are sparse, but Hu et al. found a correlation between serum trypsin levels and the severity of AP. Patients with a more severe course of disease show higher levels of serum trypsin [28]. In contrast, our clinical trial showed an inverse correlation of serum trypsin concentration with severity of AP. One possible explanation might be that blood samples were collected on admission, whereas Hu et al. collected blood samples at different, undisclosed timepoints. Another potential cause for the difference in trypsin levels might be the difference in etiology of AP between the two studies; 33% of the patients with AP included in our publication but none in the publication by Hu et al. presented with AP due to alcohol consumption.

Our systematic review as well as our clinical trial show the clinical APACHE II score to be superior to the serum markers assessed. The routine use of trypsinogen-2, trypsinogen-1, TAP or trypsin for predicting the severity of AP cannot be recommended. As some studies showed higher concentrations of TAP and trypsin in patients with MAP, and others in patients with SAP, further research in the form of a large controlled randomized prospective trial is needed.

## Figures and Tables

**Figure 1 life-14-01055-f001:**
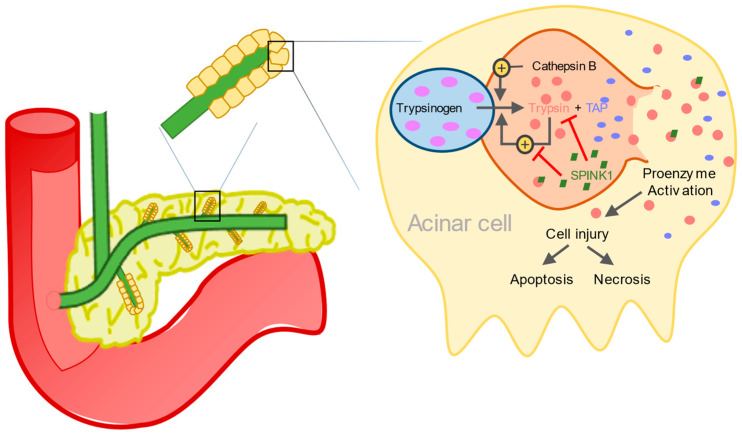
Activating and inhibiting factors in the cascade of trypsin activation in AP in the acinar cell of the pancreas.

**Figure 2 life-14-01055-f002:**
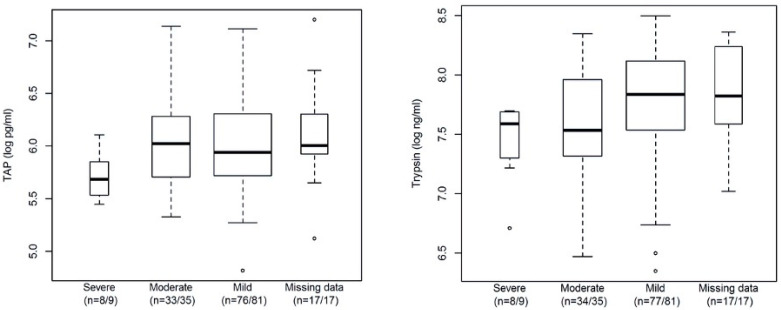
Biomarker levels at study inclusion among patients with severe, moderate and mild acute pancreatitis according to the Atlanta 2012 criteria as well as among those with missing data. The boxes are drawn with widths proportional to the square root of the number of observations in the four groups.

**Figure 3 life-14-01055-f003:**
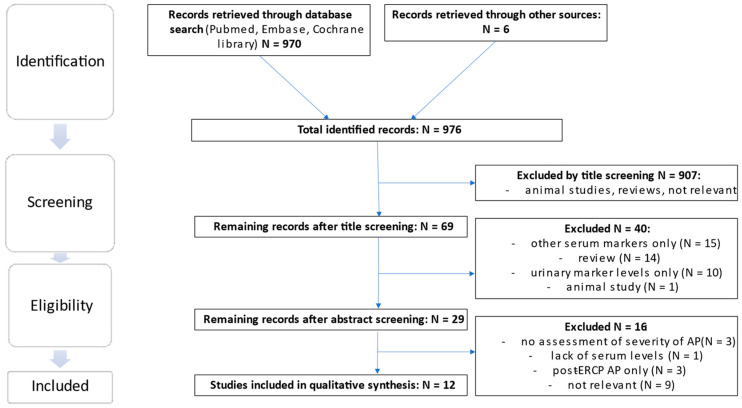
Flow chart of study selection for inclusion in systematic literature review.

**Table 1 life-14-01055-t001:** Patient characteristics.

**Characteristic**	**All Patients** **(*n* = 142)**	**Severe** **(*n* = 9)**	**Moderate** **(*n* = 35)**	**Mild** **(*n* = 81)**	**Missing Data** **(*n* = 17)**
Age, years	57 (44, 72)	68 (45, 70)	62 (52, 73)	55 (39, 73)	51 (34, 68)
Female, *n* (%)	61 (43)	3 (33)	8 (23)	42 (52)	8 (47)
Pregnancy, *n* (%)	1 (1)	0 (0)	0 (0)	1 (1)	0 (0)
Cause of pancreatitis, *n* (%)					
Alcoholic	33 (23)	5 (56)	11 (31)	13 (16)	4 (24)
Biliary	86 (61)	4 (44)	17 (49)	55 (68)	10 (59)
Other	23 (16)	0 (0)	7 (20)	13 (16)	3 (18)
History of pancreatitis, *n* (%)	26 (18)	0 (0)	10 (29)	13 (16)	3 (18)
History of rheumatic disease, *n* (%)	9 (6)	0 (0)	1 (3)	6 (7)	2 (12)
CT scan, *n* (%)	58 (41)	6 (67)	24 (69)	27 (33)	1 (6)
Time from study inclusion to CT scan, days	0	0	0	0	0
	(0, 2)	(0, 1)	(0, 2)	(0, 2)	(0, 0)
Findings CT scan ^a^, *n* (%)					
Normal	7 (5)	0 (0)	3 (9)	4 (5)	0 (0)
Edematous pancreatitis	48 (34)	5 (56)	19 (54)	23 (28)	1 (6)
Pancreatic necrosis	10 (7)	4 (44)	6 (17)	0 (0)	0 (0)
Abscess	0 (0)	0 (0)	0 (0)	0 (0)	0 (0)
Pseudocyst	5 (4)	0 (0)	5 (14)	0 (0)	0 (0)
Bleeding	2 (1)	0 (0)	2 (6)	0 (0)	0 (0)
ERCP ^b^, *n* (%)	18 (21)	1 (25)	3 (18)	13 (24)	1 (10)
Time from study inclusion to ERCP ^b^, days	2	21	2	2	3
	(1, 4)	(21, 21)	(1, 10)	(1, 3)	(3, 3)
	**Main Analysis**				
**Characteristic**	**All Patients (*n* = 136)**		**Organ Failure or Death (*n* = 29)**		**No Organ Failure or Death (*n* = 107)**
Age, years	57 (45, 72)		65 (49, 76)		55 (40, 72)
Female sex, *n* (%)	56 (41)		8 (28)		48 (45)
Cause of pancreatitis, *n* (%)					
Alcoholic	33 (24)		11 (38)		22 (21)
Biliary	83 (61)		15 (52)		68 (64)
Other	20 (15)		3 (10)		17 (16)
Trypsin ^c^, ng/mL	2292 (1699, 3046)		1888 (1600, 2350)		2459 (1815, 3317)
TAP ^d^, pg/mL	381 (298, 534)		348 (292, 448)		382 (301, 538)
APACHE II ^f^ (score)	6 (4, 8)		9 (6, 14)		5 (4, 8)

Abbreviations: CT, computed tomography; ERCP, endoscopic retrograde cholangiopancreatography; APACHE II, Acute Physiology and Chronic Health Evaluation II. Data are median (interquartile range) if not stated otherwise. ^a^ Multiple findings per patient possible. ^b^ Among patients with biliary pancreatitis. ^c^ All patients—available in 27/29 and 103/107 patients with and without organ failure or death, respectively; patients without organ failure at study inclusion—available in. ^d^ All patients—available in 26/29 and 102/107 patients with and without organ failure or death, respectively; patients without organ failure at study inclusion—available in.

**Table 2 life-14-01055-t002:** Ratios of geometric means (with 95% confidence intervals) of biomarker levels at the time of inclusion in the study; comparisons between patient subgroups defined according to the Atlanta 2012 classification of acute pancreatitis.

	**Main Analysis**		**Sensitivity Analysis**	
**Severe (*n* = 9)** **Moderate (*n* = 35)** **Mild (*n* = 81)**	**Ratio of Geometric Means (95% CI)**	***p*-Value**	**Ratio of Geometric Means (95% CI)**	***p*-Value**
**Trypsin** ^a^				
Severe vs. mild	0.72 (0.51, 1.00)	0.053	0.77 (0.55, 1.07)	0.115
Severe vs. moderate	0.90 (0.63, 1.28)	0.550	1.00 (0.71, 1.41)	0.989
Moderate vs. mild	0.80 (0.66, 0.96)	0.019	0.77 (0.65, 0.91)	0.003
**TAP** ^b^				
Severe vs. mild	0.74 (0.54, 1.01)	0.055	0.73 (0.54, 0.98)	0.035
Severe vs. moderate	0.72 (0.52, 1.00)	0.051	0.72 (0.53, 0.99)	0.042
Moderate vs. mild	1.02 (0.86, 1.22)	0.794	1.01 (0.85, 1.19)	0.948
	**All Patients (*n* = 136)**			
**Prognostic Variable**	**N**	**OR (95% CI)**	** *p* **	**AUC**
**Trypsin** **(by one SD increase)**	130	0.64 (0.41, 0.97)	0.036	0.66
**TAP** **(by one SD increase)**	128	0.88 (0.56, 1.36)	0.575	0.55
**APACHE II** **(by one SD increase)**	134	2.76 (1.75, 4.70)	<0.001	0.73

Abbreviations: CI, confidence interval. ^a^ Main analysis—available in 8/9 (89%), 34/35 (97%) and 77/81 (95%) patients with severe, moderate and mild AP; sensitivity analysis—available in 7/9 (78%), 34/35 (97%) and 75/81 (93%) patients with severe, moderate and mild AP after the removal of three outliers with an unusually low trypsin level (one from the severe AP group and two from the mild AP group). ^b^ Main analysis—available in 8/9 (89%), 33/35 (94%) and 76/81 (94%) patients with severe, moderate and mild AP; sensitivity analysis—available in 8/9 (89%), 33/35 (94%) and 75/81 (93%) patients with severe, moderate and mild AP after the removal of one outlier with an unusually low TAP level (from the mild AP group).

**Table 3 life-14-01055-t003:** Univariate logistic models for the prediction of organ failure or death within four days of inclusion in the study.

	All Patients (*n* = 136)			
Prognostic Variable	N	OR (95% CI)	*p*	AUC
**Trypsin** **(by one SD increase)**	130	0.64 (0.41, 0.97)	0.036	0.66
**TAP** **(by one SD increase)**	128	0.88 (0.56, 1.36)	0.575	0.55
**APACHE II** **(by one SD increase)**	134	2.76 (1.75, 4.70)	<0.001	0.73

Abbreviations: OR, odds ratio; CI, confidence interval; AUC, area under the ROC curve; SD, standard deviation; APACHE II, Acute Physiology and Chronic Health Evaluation II. Biomarker levels were log-transformed before all analyses to normalize their distribution.

**Table 4 life-14-01055-t004:** Area under the curve (AUC) values measured by receiver-operating characteristics (ROC) curve analysis for the prognosis of AP (severe vs. mild) and median serum concentrations of trypsinogen-2 and trypsinogen-1.

Publication	Year	AUC	Cut-Off (ug/L)	N (AP)	N (SAP)	Median Conc. SAP (ug/L)	N (MAP)	Median Conc. MAP (ug/L)	Time of Sample Collection	Definition Used for Severity Used
**Trypsinogen-2**										
Hedström J	2001	0.792	911	64	21	1829	43	768	on admission	Atlanta 1992
Lempinen M	2003	0.745	-	64	19	2930	45	1890	on admission	Atlanta 1992
Hedström J	1996	0.744	-	110	28	2251	82	992	0–48 h	According to clinical outcome
Rainio M	2019	0.726	1512	239	67	1186	172	254	0–12 h	Atlanta 2012
Hedström J	1996	0.721	-	59	19	1800	40	1000	0–24 h	Major local or systemic complication
Regnér S	2008	-	-	140	16	1334	124	704	0–24 h	Atlanta 1992
Kemppainen E	2000	-	-	92	73	2880	19	923	on admission	Atlanta 1992
**Trypsinogen-1**										
Lempinen M	2003	0.768		64	19	350	45	220	on admission	Atlanta 1992
Rainio M	2019	0.656	1279	239	67	451	172	316	0–12 h	Atlanta 2012

**Table 5 life-14-01055-t005:** Area under the curve (AUC) values measured by receiver-operating characteristics (ROC) curve analysis for the prognosis of AP (severe vs. mild) and median serum concentrations of TAP.

Publication	Year	AUC	N (AP)	N (SAP)	Median Conc. (SAP) (nmol/L)	N (MAP)	Median Conc. (MAP) (nmol/L)	Time of Sample Collection	Definition Used for Severity
**Allemann A (Present study)**	**2024**	**0.55**	**128**	**26**	**-**	**102**		**On admission**	**Organ failure or death**
**Lempinen M**	2003	0.823	64	19	5	45	1	on admission	Atlanta 1992
**Mayer JM**	2000	-	25	16	1.6	9	1	on admission	Atlanta 1992
**Kemppainen E**	2001	-	172	35	-	137	-	0–6 h	Atlanta 1992
**Pezzilli R**	2004	0.253	34	12	-	22	-	on admission	Atlanta 1992

**Table 6 life-14-01055-t006:** Median serum concentrations of trypsin in SAP vs. MAP.

Publication	Year	AUC	N (AP)	N (SAP)	Median Conc. (SAP)	N (MAP)	Median Conc. (MAP)	Time of Sample Collection	Definition Used for Severity
**Allemann A (Present study)**	**2024**	**0.66**		**27**	**-**	**103**	**-**	**On admission**	**Organ failure or death**
**Hu J**	2018	-	140	94	71–124 nmol/L	46	65 nmol/L	not disclosed	Not disclosed

## Data Availability

The data of the single center observational study as well as the literature review will be made available upon reasonable request.

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
