# Peer review of "Trypsin and Trypsinogen Activation Peptide in the Prediction of Severity of Acute Pancreatitis"

_life, 2024, doi:10.3390/life14091055_

Round 1

Reviewer 1 Report

Comments and Suggestions for Authors

The article “Trypsin and Trypsinogen Activation Peptide in the Prediction of Severity of Acute Pancreatitis” reviews recent assessments concerning predictive value of serum trypsin and Trypsinogen Activation Peptide (TAP) for the severity of acute pancreatitis (AP). The topic is very relevant, because it is very important to predict the severe course in AP as early as possible. A variety of scores have been developed in the past to predict or assess the severity of AP, however, current predictive models lack specificity and reliability, especially in the early course of disease. This review showed promising results for the predictive value of serum trypsinogen-2.

This study presents a systematic review of literature regarding the correlation of serum trypsinogen, trypsin and TAP with severity of AP, as well as the assessments of the utility of trypsin and TAP levels as predictors of severity of disease in serum of patients admitted for AP, by means of a post-hoc analysis of prospectively collected data of a single center cohort. The literature search started from yielded 970 result and the post-hoc analysis of a prospective, single center, observational cohort-study included 142 patients.

The manuscript is well organised; however, the disadvantage could be that there is no comparative analysis with the control group. In particular, this could have been done by performing the post-hoc analysis. So, I would recommend adding comparative data obtained with the control group of patients without AP diagnosis.

The literature review covering 35 sources has been provided.

The material for the publication is presented in good English, thus I do recommend this article for publication in Life.

Author Response

The authors present a well organised manuscript on a relevant topic.

  1. The disadvantage is that there is no comparative analysis with the control group. The addition of comparative data obtained with the control group of patients without AP diagnosis is suggested.

Our Reply: The study does not include a control group of patients without AP, as patients without AP do not show detectable serum concentration of TAP (<1ng/l) (A. Barassi, 2010), and normal serum levels of trypsin (115-350 ng/l) (Heinrich H.C., 1979).

Therefore, in this study what could be considered a control group would be the patients with mild AP (vs severe AP), which is reported in this paper.

Reviewer 2 Report

Comments and Suggestions for Authors

I do not understand what the authors intended to prove with this study. It is a combination of a systematic review and some personal data but in the discussions, data are not so complementary. For example, the authors talk about the use of tripsyne and TAP for predicting the severity of AP in their study and about the use of trypsinogen 2 in the review. 

I suggest 2 ways: either split the article into 2 distinctive parts or make a direct correlation between their own study and literature review.

Author Response

I do not understand what the authors intended to prove with this study. It is a combination of a systematic review and some personal data but in the discussions, data are not so complementary. For example, the authors talk about the use of tripsyne and TAP for predicting the severity of AP in their study and about the use of trypsinogen 2 in the review.

I suggest 2 ways: either split the article into 2 distinctive parts or make a direct correlation between their own study and literature review.

Our Reply: We agree with the reviewer and have thus reorganized the article to separate the clinical study more clearly from the literature review.

Since the three markers included in the review are interconnected by cleavage of trypsinogen-2 into trypsin and TAP, we decided to include trypsinogen-2 in the review part of the paper for comparison and to provide further context for the reader.

Reviewer 3 Report

Comments and Suggestions for Authors

The paper is divided into two: systematic review and single study. This causes a bit of confusion. Perhaps it is better to describe the individual study and compare it on what differs in the literature. I proposed a paper set as a single study versus literature.

The paper reports data from three biomarkers to predict acute pancreatitis. The authors retrieve data from a meta-analysis and their own single study. The data from the single study are interesting and very complete. To facilitate the reading and understanding of all the data (literature and single study) I have some suggestions: evaluate all the data of the single study and compare them with the data present in the literature. Currently evaluating the tables present in the table it is not clear how the data of the single study blend with those present in the literature. It is not clear whether this paper should be considered a meta-analysis review or an original paper. Furthermore, three figures of Roc curves can be made (one for each marker) and in each there can be different Roc curves with the best cut-off with sensitivity and specificity based on the data present in the various paper in the literature. I strongly recommend that you also include the data from the individual study in this figure. From reading this paper we must have a conclusion on how to diagnose acute pancreatitis with the three biomarkers and with what cut-off values. The data present in this paper , therefore, are interesting, they just require a different organization. Tables are designed as a list of data, when they should make you perceive how the data is different. Another suggestion is to merge similar data by inserting multiple references to that data. In the discussions it is also interesting to underline what are the best methods for the dosages of these three markers to best diagnose acute pancreatitis and its monitoring. Comments on the Quality of English Language

Make the text flow more smoothly. Make simple, short sentences

Author Response

The paper is divided into two: systematic review and single study, which is confusing. The data from the single study are interesting and very complete.

  1. It is suggested to describe the individual study and compare it on what differs in the literature; as a single study versus literature.

Our Reply: Thank you very much for this excellent suggestion; we agree and changed the structure of our paper.

  1. The data as presented in the tables of the single study does not indicate clearly how it blends with those present in the literature.

Our Reply: Thank you for your comment. We agree with the reviewer and have included our findings in the review tables for a direct comparison with previously reported values in the literature to enhance readability.

  1. It is not clear whether this paper should be considered a meta-analysis review or an original paper.

Our Reply: Thank you for this comment. We consider this an original paper with a comparative literature review and not a meta-analysis as we did not include the necessary statistical tests to be allowed to use this term. Due to the relative scarcity and heterogeneity of data, we opted to go for a comparative review and not a meta-analysis.

  1. It is suggested the authors make three figures of ROC curves (one for each marker) indicating the best cut-off values with sensitivity and specificity based on the data present in the various papers in the literature.

Our Reply: Thank you for this valuable comment. We have added a figure with ROC curves as a supplemental figure, with further laboratory values tested in the same instance to give the reader an understanding of the relative value compared to other tested biomarkers. Due to the relatively low AUC values which were not in range of clinical relevance we did not offer such a cut-off value as this would not reflect a clinically relevant value.

  1. The lecture of this article should lead to the clear conclusion on how to diagnose acute pancreatitis with the three biomarkers and with what cut-off values.

Our Reply: Currently CRP-assessment and APACHE II score remain the gold standard for diagnosis of severe AP, since it has high specificity and sensitivity and is readily available. The data of our clinical study does not lead to conclusive cut off values for diagnosis of severe AP applicable to clinical practice and we have therefore opted to not include a cut off value as such a value would not represent a clinically meaningful figure.

  1. It is suggested to merge similar data into the tables of the single study by inserting multiple references to that data.

Our Reply: Thank you for this suggestion. We included our own results in the tables of the literature review to facilitate comparison.

  1. It would be beneficial to underline what the best methods for the dosages of the three markers are to best diagnose acute pancreatitis and for monitoring the course of disease.

Our Reply: Thank you for this valuable comment. As per our statement above we have opted to not include a cut-off value as this would not represent a clinically meaningful value. The aim of this study was to provide information about the value of the studied biomarkers as an early marker of severity, and the study was not designed along the lines of disease monitoring. Clinical experience would dictate that markers such as CRP would likely be better candidates to fulfil this role, as they allow for a reflection of inflammatory processes in the body as compared to specifical pancreatic insult, for which our studied biomarkers were selected in this study.

  1. It is advised to make the text flow more smoothly and make simpler, shorter sentences.

Our Reply: Thank you for this valuable comment, we have reviewed the manuscript and improved the language.

Round 2

Reviewer 2 Report

Comments and Suggestions for Authors

I appreciate the effort you made to improve the manuscript. It looks better now!

Author Response

Thank you dear reviewer for your feedback and effort in order to improve our manuscript.

Reviewer 3 Report

Comments and Suggestions for Authors

the authors modified the work based on the indications received. I notice that there are still small things to change. in the structure of the work there are two times the title results. it is necessary to standardize the work in introduction, material and method, discussion without having duplications. The comparison between the study and the rest is not yet clear from the tables.

Author Response

Dear reviewer, thank you for taking the time to review our manuscript. We have amended the points as per your inputs, and removed the duplicates in titles. We have furthermore highlighted our results in the tables for comparison with previous work to improve readability.

We hope that these amendments are to the reviewers satisfaction.

Kind regards on behalf of the authors

Sebastian M. Staubli